# The QQUIC Transport Protocol: Quantum-Assisted UDP Internet Connections

**DOI:** 10.3390/e24101488

**Published:** 2022-10-18

**Authors:** Peng Yan, Nengkun Yu

**Affiliations:** Centre for Quantum Software and Information, Faculty of Engineering and Information Technology, University of Technology Sydney, Sydney 2007, Australia

**Keywords:** transport protocol, QUIC, QKD, handshake, key exchange

## Abstract

Quantum key distribution, initialized in 1984, is a commercialized secure communication method that enables two parties to produce a shared random secret key using quantum mechanics. We propose a QQUIC (Quantum-assisted Quick UDP Internet Connections) transport protocol, which modifies the well-known QUIC transport protocol by employing quantum key distribution instead of the original classical algorithms in the key exchange stage. Due to the provable security of quantum key distribution, the security of the QQUIC key does not depend on computational assumptions. It is possible that, surprisingly, QQUIC can reduce network latency in some circumstances even compared with QUIC. To achieve this, the attached quantum connections are used as the dedicated lines for key generation.

## 1. Introduction

The problem of generating a secret key between two remote parties affects anyone wishing to use encryption in modern communication. Traditionally, symmetric encryption suffers one enormous shortcoming, it is necessary for either the sender or the recipient to create a key and then send it to the other party. While the key is in transit, it could be stolen or copied by a third party, who can then destroy the encryption. In the 1970s, the Diffie–Hellman key agreement method [1], a method of distributing keys without actually sending the keys themselves, was developed. It provides the basis for a variety of authenticated protocols. For example, Transport Layer Security (TLS) protocol uses the ephemeral Diffie–Hellman to enable forward secrecy.

In modern Internet engineering, significant efforts have been focused on establishing more secure connections by employing and updating TLS protocols. Deployed in 2012 by Google [2], QUIC (pronounced “quick”) is a new general-purpose transport layer network protocol based on UDP to improve the performance of connection-oriented web applications based on the Transmission Control Protocol (TCP) with greatly reduced latency. Unlike TLS over TCP, QUIC sends data as stream frames carried in the QUIC packets by its reconstructed transport layer rather than through the TLS record layer directly. The TLS and QUIC protocols work cooperatively, where TLS mainly provides the TLS handshake and state change messages for the authentication and negotiation of parameters. Nowadays, QUIC is widely used by more than half of all connections within the Chrome web browsers, especially for gaming, streaming media, and VoIP services.

However, both the RSA and Elliptic Curve Diffie–Hellman asymmetric algorithms that set up the TLS exchange are vulnerable to quantum computers big enough to run Shor’s algorithm. For instance, the Diffie–Hellman key agreement itself is a non-authenticated key-agreement protocol. In other words, the security of the Diffie–Hellman key agreement method, as well as RSA, is threatened by Shor’s polynomial-time algorithm for the discrete logarithm problem [3], which can be performed on emergent quantum computers [4,5,6,7]. Shor’s algorithm can cause modern asymmetric cryptography to collapse since it is based on large prime integer factorization or the discrete logarithm problem. Therefore, they are not considered to be quantum-safe or post-quantum cryptographic algorithms.

Using quantum mechanics, Bennett and Brassard introduced a famous quantum key distribution (QKD) method named BB84 [8,9] in 1984, which enables two remote end nodes to establish provably secure random keys. To implement this quantum key distribution, requires quantum encoders and decoders to be capable of preparing and measuring weak coherent quantum states. In [10], the world’s first network, the DARPA Quantum Network, was built by BBN, Harvard, and Boston University, which delivered end-to-end network security via high-speed QKD. Another QKD network [11] designed and implemented by the European project SEcure COmmunication based on Quantum Cryptography (SECOQC) was put into operation in Vienna in 2008. Building the 1200-mile quantum communication landline between Beijing and Shanghai in 2016 and the quantum communication satellite (known as Micius) in 2017, China has the world’s first space–ground quantum network [12]. Recently, many research teams have succeeded in building and operating more practical quantum cryptographic devices [13,14,15,16].

To the best of our knowledge, QKD has not been employed as a component in the design and implementation of modern practical Internet protocols. This is very surprising regarding the fundamental role played by key distribution in communication security. In this paper, we propose the QQUIC transport protocol by carefully adding QKD into the TLS handshake in the well-known QUIC protocol to enhance the generation of shared secrets. We develop an encrypted transport protocol to improve the security and transport performance for HTTPS traffic based on the new TLS 1.3 and quantum key distribution BB84. The basic motivation is that quantum key distribution (QKD) can generate a completely random secret known only to the communication sides, which can be used to improve security significantly and optimize the procedure of key exchange in the conventional transport protocol. The QKD is mainly used to replace the key exchange to create the shared secret in the initial handshake, and the security of the application data after the connection establishment is still undertaken by the TLS. Interestingly, QQUIC can reduce network latency in some circumstances even compared with QUIC.

## 2. Preliminaries

### 2.1. Quick UDP Internet Connection

Transport Layer Security (TLS) is designed to achieve the goal of a safe connection between a client and a server by providing privacy, integrity, and authentication for the data transmitted over the network. The security of data can be guaranteed via symmetric cryptography with a known shared key, and public-key cryptography can be used to provide the digital signature and authenticate the identity of the communicating parties, and the message authentication code (MAC) generated for every message can be used to detect message tampering and forgery.

Designed in 2012 by Google, QUIC [2] has undergone great improvements in several aspects, such as stream multiplexing, flow/congestion control, loss recovery, and connection migration. The most significant advantage of QUIC is that it has much lower connection establishment latency compared with the combination of TCP and TLS. QUIC employs the TLS handshake to drive the keys used for data encryption and also relies on TLS for authentication, parameter negotiation, and state change information. Here, we mainly focus on the connection establishment process and briefly restate the full handshake of TLS used in the current QUIC, as shown in Figure 1.

The client sends a ClientHello(CHLO) package to initiate a new handshake. The CHLO package contains information on (a) a random client number $R_{c}$ to generate the master secret, (b) supported TLS versions and cipher suits, and (c) extensions for session resumption, signature, and compression algorithms. The server responds to the client with a ServerHello (SHLO) package at once, which also contains a random server number $R_{s}$, the choice of TLS version and cipher suits, and response for extensions. Then a signed certificate is sent by the server to prove its identity to the client, which contains the server’s long-term public key. Furthermore, a Server Key Exchange message is sent if the public key contained in the server’s certificate is not sufficient for the client to finish the key exchange. Finally, a Server Hello Done message is sent to indicate that the server is done and waiting for the client’s response.

Once the client receives the ServerHello message, it first verifies the server’s certificate and caches the information needed for a reconnection. Optionally, the client may need to send its certificate if required by the server. Similarly, a Client Key Exchange message encrypted by the server’s public key is also sent by the client. Now the client can infer the master secret based on the Key Exchange messages and the random number $R_{c}$ and $R_{s}$. A Change Cipher Spec message will be sent to notify the server that any data sent by the client from now on will be encrypted by the symmetric key derived from the master secret, followed by a Finished message, which contains a hash of all the handshake messages sent previously. Once the server uses its private key to decrypt the Client Key Exchange message, the server can also calculate the same master secrets similar to the client. Now the server can use the symmetric key also derived from the master secret to decrypt the client handshake Finished message and check the validity of all previous handshake messages. After this, the server will also send the Change Cipher Spec and Finished message similar to the client to achieve the same goal. Once the client successfully verifies the server’s Finished message, a full TLS Handshake is completed, and application data can be transmitted securely. Therefore, TLS 1.2 takes 2-RTT to create a new secure connection.

### 2.2. Quantum Key Distribution

BB84 [8] is a quantum key distribution scheme developed by Charles Bennett and Gilles Brassard in 1984. The protocol is provably secure [17], relying on the fundamental aspect of quantum mechanics that information gain is only possible at the expense of disturbing the signal [18,19]. The workflow of BB84 is shown in Figure 2.

Alice randomly generates two strings, $a=a_{0}a_{1}\cdots a_{n}$ and $b=b_{1}b_{2}\cdots b_{n}$. *a* is called the basis string. If $a_{i}=0$, she chooses a qubit in {|0〉,|1〉} according to $b_{i}$; otherwise, she chooses a qubit in {|+〉,|−〉} ($|+\rangle =\frac{|0\rangle +|1\rangle }{\sqrt{2}}$ and $|-\rangle =\frac{|0\rangle -|1\rangle }{\sqrt{2}}$) according to $b_{i}$. These qubits are sent by Alice to Bob in strict sequence.After receiving these quantum states, Bob randomly measures each qubit in basis {|0〉,|1〉} or in basis {|+〉,|−〉}, and records his choice of basis string as $a^{\prime }$. After he receives the measurement outcome $b^{\prime }$, $a^{\prime }$ is sent to Alice.Alice discards the bits of *b* in the locations where *a* and $a^{\prime }$ do not match. Alice announces *a* first; then she randomly takes half of the rest of the bits of *b* to be the checking bits $b_{c}$ and announces the selection.Bob discards the bits of $b^{\prime }$ in the locations where *a* and $a^{\prime }$ do not match after knowing *a*. Then he checks if the states of qubits in checking bits $b_{c}$ before and after the measurement are the same. If the disagreement is less than an acceptable threshold, the quantum channel is considered to be reliable, and the remaining bits can be used to obtain the shared key bits.

However, the standard BB84 protocol [8] is not practical since it is not feasible to generate strings of truly perfect single photons efficiently [20]. The decoy-state method [14,21,22,23] was proposed to use a few different photon intensities to perform almost as well as a single photon. Different configurations [24,25] of decoy states and the length of keys [26] would also affect the secret key rates. An optimal configuration would reduce the time for gathering sufficient statistics to generate a secure key, which is important to make classical and quantum channels collaborate well on the same time scale. Moreover, the high-speed state of polarization generation [27,28] and other degrees of freedom of polarization, such as time-bin [15], can also be implemented to improve the efficiency of QKD.

## 3. QQUIC Transport Protocol

In the previous section, we briefly explained how BB84 works to generate a shared secret between two sides. If we want to combine these properties in traditional transport protocols, it requires a quantum channel to generate, transport, and measure the single polarized photon in a predefined way, and it also needs two peers to communicate their basis choice and measurement results through the classical channel. This may appear to be much more complicated compared with the traditional key exchange process, but the quantum channel and classical channel can work in parallel to finish the handshake, and QKD can also continuously work to provide keys for key updates after the connection has been established.

We introduce a new secure transport protocol called QQUIC, where QKD replaces the key exchange stage in QUIC. The encryption and transmission of the application data after connection establishment works the same way as QUIC, except for the initial full handshake for a new connection and key updating. There are two channels during the connection; one is the classical channel used in traditional transport protocols, and the other quantum channel is used for quantum operations on the photons. The network structure of the protocol works as shown in Figure 3.

Each side has a quantum host and a classical host. The quantum hosts and their authenticated quantum channel are responsible for generating the shared secret to replace the classical key exchange. For the initial full handshake and the periodical key updating, the client needs to activate its quantum host to generate and send strings of special polarized photons, where the quantum host undertakes the generation of true randoms *a* and *b*, which determines the photon states. Meanwhile, the client informs the server of the configuration of the photons through the classical channel, including the state basis for photon generation and measurement, the number and sending speed of photons, etc. After the server knows the configuration of these photons, its quantum host generates a random basis string $a^{\prime }$ and uses it to get the measurement outcome $b^{\prime }$ of the received photons. After the measurement, the two sides communicate their basis choices *a* and $a^{\prime }$ and discard the useless bits in *b* and $b^{\prime }$. After verifying the validity of the quantum channel, the secure, shared secret can be created and used for key derivation. Notice that the client cannot announce its basis string $a^{\prime }$ until it knows the server has completed the measurements of the photons, which means the QKD needs at least 2-RTT to complete the shared secret negotiation. Here we give a detailed description of the workflow of the full handshake, shown in Figure 4.

Initially, the client has no information about the server, so it needs to initiate the connection via a ClientHello (CHLO) message. The CHLO message contains the lists of supported TLS versions, cipher suites, signature algorithms, and extensions for session resumption, which are similar to the CHLO message in TLS 1.3. However, the CHLO package provides no information related to key exchange, such as a random client number or key exchange extension. Instead, the client requires its quantum host to generate two private binary randoms, *a* and *b*, to determine how to generate a string of polarized photons and send them through the quantum channel at the same time, as explained in BB84. Thus, the CHLO message usually needs to contain some extra information about the photons, such as the set of state basis used for photon generation and measurement and the number and sending speed of photons. The transmission of photons in the quantum channel works in parallel with the CHLO message in the classical channel.

The server gives a ServerHello (SHLO) message as a response to the CHLO package. The SHLO message contains: (a) the server configuration information, (b) the choice of cipher suites and signature algorithms, and (c) a certificate and certificate verification. The server’s quantum host needs to receive these polarized photons in the same order and measure them in sequence by a basis string $a^{\prime }$, which denotes the random selections of two possible bases provided by the client. After this, the server keeps the measurement results $b^{\prime }$ private and sends the string $a^{\prime }$ to the client through the classical channel.

Once the client receives the SHLO message from the server, it can verify the identity of the server via the certificate and signature. After this, the client has to wait for the handshake message about the server’s choice of measurement basis $a^{\prime }$ and discard the bits in *b* where the two sides choose differently. Then the client needs to randomly take half of the remaining bits in *b* to create a random $b_{c}$, and the rest of the bits in *b* can be used to generate the shared secret. Now the client needs to send *a* (generating basis) to inform the server as to how these photons are created, and $b_{c}$ (checking basis) to let the server decide which photons are used for checking. Finally, the client should use the derived key to encrypt the Change Cipher Spec message and the Finished message, which contains a hash of all the handshake information up to this point to avoid tampering.

Once receiving the handshake messages from the client, the server should first check whether or not the quantum measurement on the photons corresponding to the bits in $b_{c}$ have changed their states, where the states before measurement can be inferred from *a* and $b_{c}$. If the difference rate is above a safe threshold, this means there is too much noise or there is an eavesdropper in the quantum channel. Thus, the handshake has failed, and it is necessary to try again. Otherwise, the server can safely derive the same shared secret based on *a* and $b_{c}$ from the client and its measurement results $b^{\prime }$. Now the server can decode the Finished message from the client to check the validity of all the previous messages. In the end, the server also sends the Change Cipher Spec and Finished message to the client. After the client checks this Finished message, a safe connection has been established.

As a comparison, the establishment of a safe connection via QKD requires 2-RTT, which is 1-RTT slower than TLS 1.3. However, if the success rate of the validity checking for the quantum channel is high enough, the client can also send some unimportant encrypted application data after the client’s Finished message in the first round trip, and the server can also directly give its response to the request after its Finished message. This “quick start” strategy can be used in most HTTP requests, where it usually begins with an unimportant request, such as permission for log-in. As for the session assumption, the PSK mode and 0-RTT reconnection used in QUIC can also be deployed here. Again, because the quantum channel needs 2-RTT to generate the shared secret safely, the application data in the first two round trips during the handshake are still encrypted by the previous old key.

It has been confirmed that a connection encrypted by one session key for a long time is not safe enough, even using authenticated encryption with associated data. To guarantee security, TLS 1.3 proposes to use the KeyUpdate message to update session keys in an existing connection, which takes 1-RTT to notify each side and derive the next shared traffic secret based on the current traffic secret and KeyUpdate message, as shown in Figure 5a.

In addition to the higher security level of the master secret derived from QKD, the most obvious advantage of deploying QKD is that the key generation and application data transmission can work in parallel. As shown in Figure 5b, after the connection has been established, the client can keep sending strings of photons continually, and each string can generate a master secret. The server measures a group of photon strings once and negotiates the generation and measurement bases in only one round trip, which can effectively save travel time. Since every photon string is independent of each other, we can guarantee the independence between these shared secrets. This ensures that each side always has several shared secrets stored in advance for a future key update. If both sides have made an agreement about the number of ordered packages for the lifetime of one session key, then the two sides can automatically switch to a new session key in the sequence without notification when the right time comes. The package number and the stream offset in QUIC are used to reorder the received packages and choose the right session key for different package chunks. This protocol will have huge advantages for long-term connections such as video conferences and big data transmission, which require frequent key updates. Furthermore, due to the higher security of QKD, the results from QKD can be used as a master secret directly, and there is no need to add more random materials to generate the final session keys compared with the HKDF used in TLS 1.3. The same goes for the frequent key updates in the long-term connection, where the session’s key derivations of different chunks are totally independent.

## 4. Discussion and Conclusions

In this paper, we propose a new protocol named QQUIC, which combines the advantages of QUIC and quantum key distribution to achieve more secure and effective key generation for the initial handshake and key updating. This protocol uses QKD to replace the classical key exchange procedure and performs a 2-RTT full handshake for a new connection, and with a 1-RTT time cost with the “quick start” technique. The concurrent work of the quantum and classical channels makes the key generation much more efficient and resource-saving, which has huge advantages in long-term connection applications, such as video meetings and database sharing. For implementation of QKD with a relatively high key rate, the portion of asymmetric cryptography in the network protocol may be significantly reduced. This would provide a considerable computational speedup due to the well-known fact that public-key cryptography, or asymmetric cryptography, is much slower than symmetric cryptography. Today QKD is a relatively mature technology, and some companies such as ID Quantique, QuantumCTek, and ThinkQuantum can provide feasible turn-key solutions for implementing and testing our proposal.

## Figures and Tables

**Figure 1 entropy-24-01488-f001:**
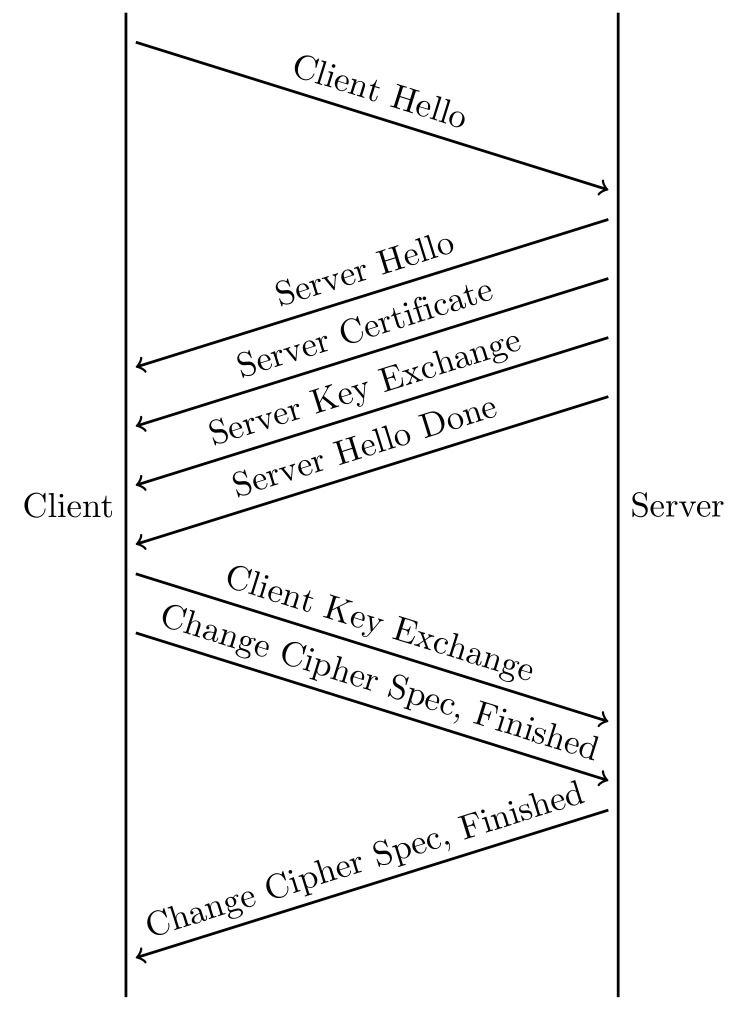
TLS 1.2 full handshake.

**Figure 2 entropy-24-01488-f002:**
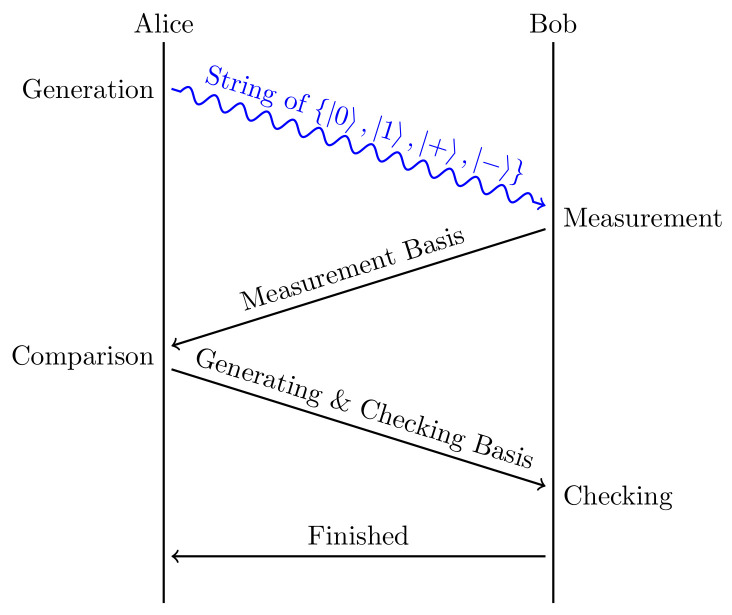
BB84 workflow. A string of polarized photons randomly in {|0〉,|1〉,|+〉,|−〉} states are sent by Alice to let Bob give random measurements in sequence. After the measurements on the photons and basis communication, both sides can generate a shared secret.

**Figure 3 entropy-24-01488-f003:**
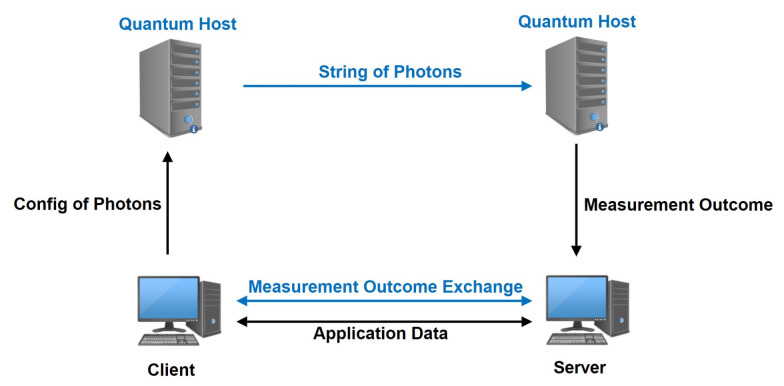
Network Structure of the QQUIC.

**Figure 4 entropy-24-01488-f004:**
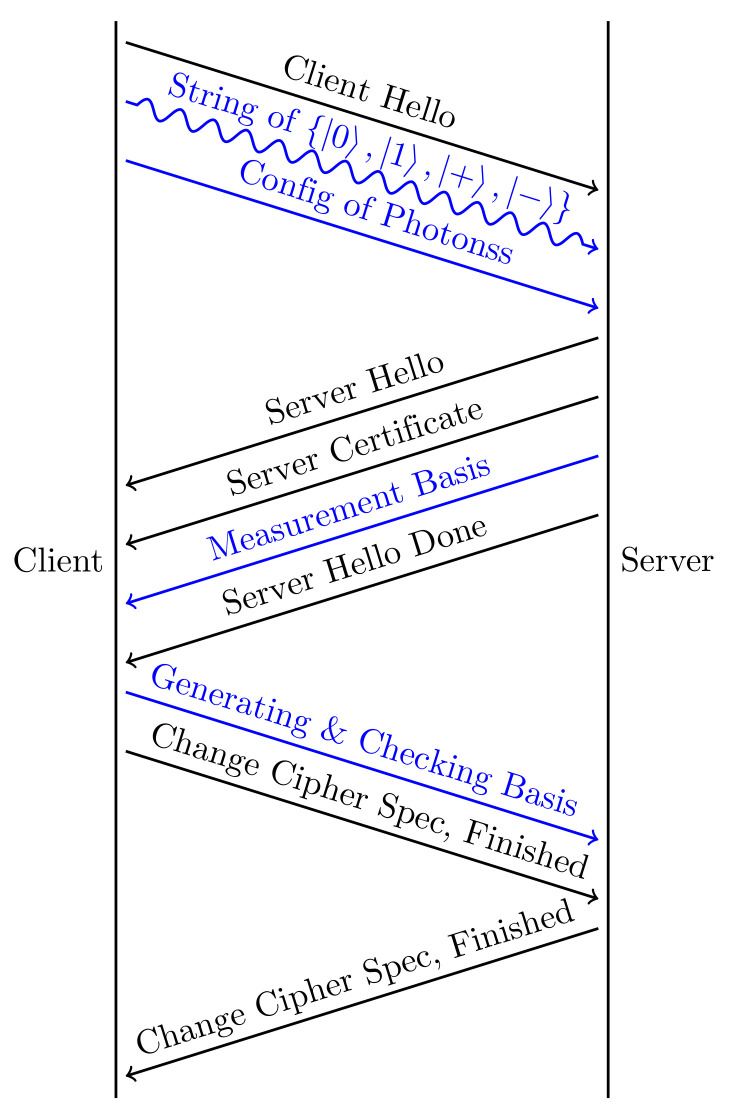
QQUIC full handshake.

**Figure 5 entropy-24-01488-f005:**
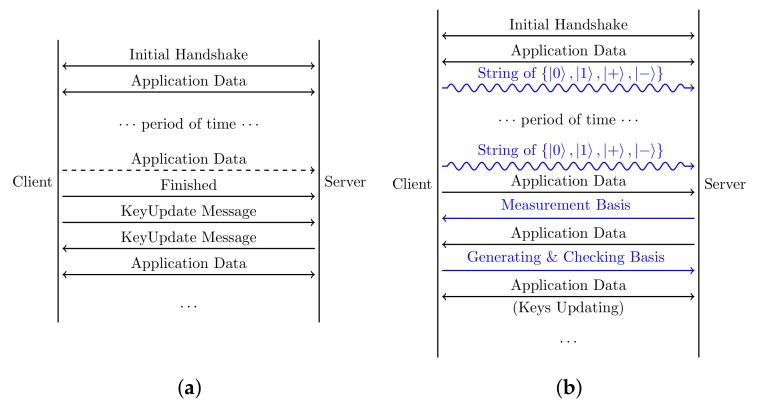
KeyUpdate in a long-term connection. (**a**) KeyUpdate in TLS 1.3; (**b**) KeyUpdate in QQUIC.

## Data Availability

Not applicable.

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
