# Peer review of "The QQUIC Transport Protocol: Quantum-Assisted UDP Internet Connections"

_entropy, 2022, doi:10.3390/e24101488_

Round 1
Reviewer 1 Report
This paper develops a new transport protocol called Quantum assisted Quick UDP internet connections (QQUIC). It is a modification of the existing QUIC transport protocol based on the quantum key distribution. The security of the proposed new protocol does not depend on the computational assumptions. The proposed protocol can also reduce network latency in certain situation. The paper is well-written and it has a logical explanation about the proposed protocol. I suggest to accept the paper after the following question is addressed: Is it possible to justify the security and effective key generation for the initial handshake and key updating of the proposed new protocol in a rigorous way? Section 3 contains a description of the proposed protocol. It would be nice to justify the security and effectiveness mathematically; or at least, clarify the reason why it is more secure and effective in a more transparent way.
Author Response
Thanks for your review very much. As we have argued in Sec 3, we propose to use QKD to replace the key exchange stage in QUIC. Compared with the traditional key exchange in QUIC, the security is naturally guaranteed by the no-cloning theorem and a reliable public classical channel. As for the effectiveness of our protocol, the collaboration between the quantum and classical channels can have a 2-RTT full handshake for a new connection and a 1-RTT time cost with the "quick start" technique. As far as we know, the QUIC protocol can be simulated in classical model suites like INET, and some early access tools like QKD Simulator and QKDNetSim could provide some basic testing of the properties of customized QKD protocols. Since we are currently unfamiliar with these tool suites, it is not feasible that we could give rigorous numerical simulations to demonstrate the effectiveness of our protocol transparently.
Reviewer 2 Report
In the submitted manuscript the authors introduce a version of the Quick UDP Internet connection (QUIC) protocol that exploits Quantum Key Distribution to improve security of the key exchange phase. The QUIC protocol is of particular importance for internet connections, since, as reported by the authors, half of all connections within the Chrome web browser employ it. In general, I find it very useful for the growth of the Quantum Communication community to have interdisciplinary proposals such as the one presented here because it showcases the characteristics and potential uses of quantum protocols such as QKD in realistic scenarios. For this reason I can support publication of this work in Entropy, provided that the authors respond to the following concerns:
1) In the introduction, the authors mention "the famous TLS's ephemeral modes". It is not fully clear to me what is meant by this phrase.
2) The authors failed to define the acronym for TCP.
3) The authors mention "emergent Quantum Computers" when describing Shor's algorithm and the perils it poses to RSA and others. I would recommend the authors to add to relevant citations. I would propose (but feel free to modify): a) T. D. Ladd, F. Jelezko, R. Laflamme, Y. Nakamura, C. Monroe, and J. L. O’Brien, “Quantum computers,” Nature 464, 45–53 (2010). b) F. Flamini, N. Spagnolo, and F. Sciarrino, “Photonic quantum information processing: a review,” Rep. Prog. Phys. 82, 016001 (2018). c) L. M. K. Vandersypen, M. Steffen, G. Breyta, C. S. Yannoni, M. H. Sherwood, and I. L. Chuang, “Experimental realization of Shor’s quantum factoring algorithm using nuclear magnetic resonance,” Nature 414, 883–887 (2001). d) A. Politi, J. C. F. Matthews, and J. L. O’Brien, “Shor’s quantum factoring algorithm on a photonic chip,” Science 325, 1221 (2009).
4) When introducing QKD, the authors mention that "quantum processors" are required. I dislike this terminology. QKD requires a Quantum State Encoder and a Quantum State Decoders. Both of these devices have less stringent requirements with respect to a quantum processor. In fact, fully operational QKD systems are already commercially available whereas Quantum Computers that requires quantum processors are further behind in development.
5) The works that are cited by the authors regarding QKD are correct and valid but I find most of them to be a bit outdated. I would recommend to enhance the list with (but feel free to modify): a) S. Pirandola, U. L. Andersen, L. Banchi, M. Berta, D. Bunandar, R. Colbeck, D. Englund, T. Gehring, C. Lupo, C. Ottaviani, J. L. Pereira, M. Razavi, J. Shamsul Shaari, M. Tomamichel, V. C. Usenko, G. Vallone, P. Villoresi, and P. Wallden, “Advances in quantum cryptography,” Adv. Opt. Photonics 12, 1012 (2020). [modern QKD review] b) A. Boaron, G. Boso, D. Rusca, C. Vulliez, C. Autebert, M. Caloz, M. Perrenoud, G. Gras, F. Bussières, M.-J. Li, D. Nolan, A. Martin, and H. Zbinden, “Secure quantum key distribution over 421 km of optical fiber,” Phys. Rev. Lett. 121, 190502 (2018). [record long fiber link for BB84] c) N. T. Islam, C. C. W. Lim, C. Cahall, J. Kim, and D. J. Gauthier, “Provably secure and high-rate quantum key distribution with time-bin qudits,” Sci. Adv. 3, e1701491 (2017). [record high secure key rate]. d) C. Agnesi, M. Avesani, L. Calderaro, A. Stanco, G. Foletto, M. Zahidy, A. Scriminich, F. Vedovato, G. Vallone, and P. Villoresi, "Simple quantum key distribution with qubit-based synchronization and a self-compensating polarization encoder," Optica 7, 284 (2020) [record low Quantum Bit Error Rate].
6) In section 2.2, a discussion about the decoy state method would be relevant because it enables the use of attenuated light sources instead of requiring true single photon states (see G. Foletto, F. Picciariello, C. Agnesi, P. Villoresi, and G. Vallone, "Security bounds for decoy-state quantum key distribution with arbitrary photon-number statistics," Phys. Rev. A 105, 012603 (2022). Also the authors should mention that QKD can be implemented with polarization as well as other degrees of freedom such as time-bin (see the previous QKD article recommendations for examples). In this regard, when the authors write "generate and send strings of special polarized photons", I would recommend replacing this with "adequately encoded qubits".
7) In section 3 of the manuscript the authors write that "the client informs the server of the configuration of the photons through the classical channel". What does the configuration of the photons mean? I believe that they mean information such as the set of state basis used for photon generation and measurement, and the number and sending speed of photons. If so, I would recommend to clarify this paragraph to prevent confusion with secret binary strings a and b used to encode the photons.
7) In figure 4 there are some typos in the spelling of the word "Basis". Please correct.
8)Modern QKD implementations use finite-key analysis to generate the secure key (see D. Rusca, A. Boaron, F. Grünenfelder, A. Martin, and H. Zbinden, “Finite-key analysis for the 1-decoy state QKD protocol,” Appl. Phys. Lett. 112, 171104 (2018)). This means that a minimum integration time is required to obtain sufficient statistics and generate a secure key. It would be useful if the authors considered this effect when estimating the required time used by the protocol.
9)In the conclusions, I would recommend for the authors to mention that QKD is a mature technology and that several companies offer turn-key solutions (ID Quantique, QuantumCTek, ThinkQuantum), and that implementing and testing these proposals can be done with today's technologies.
Author Response
We thank the viewer very much for the detailed suggestions. Here are the responses point-by-point.
1. "ephemeral modes" denotes the ephemeral Diffie-Hellman, in which a temporary DH key is generated for every connection, and the same key is never used twice. Here we mean the forward secrecy is achieved by the ephemeral Diffie-Hellman rather than its static version. We will rephrase it for clarity.
2. We will add the definition of the acronym for TCP.
3. We will add these suggested references for Shor's algorithm and its challenges to RSA and other traditional methods.
4. We agree with your comments about quantum processors; the state encoder/decoders are currently sufficient enough for operational QKD systems. We will change the terminology to give an accurate presentation.
5. Thanks for your kind suggestion. We will update the citations of QKD.
6. Just like the "quantum processors" issue mentioned before, requiring true single photons to conduct QKD is an unnecessarily strict requirement, and the term "adequately encoded qubits" would be proper and sufficient. We would add some comments about the decoy-state method and the extra degree of freedom on the implementation of polarization.
7. We would add details about the configurations of the photons and fix typos in the figures.
8. Since our work does not focus on the comparison between different configurations of decoy states in QKD, we are not sure whether it is necessary to take into account finite size effects. We would add a comment for the effect of finite-key in sec 2.2, where the integration time may become a problem if the quantum and classical channels can not collaborate well on the same time scale.
9. We would add some comments on the current technologies that can be used for testing our proposal.